# HEADLESS LANGUAGE MODELS: LEARNING WITHOUT PREDICTING WITH CONTRASTIVE WEIGHT TYING

**Nathan Godey**[1,2]    **Éric de la Clergerie**[1]    **Benoît Sagot**[1]

[1]Inria, Paris, France
[2]Sorbonne Université, Paris, France
{nathan.godey,eric.de_la_clergerie,benoit.sagot}@inria.fr

## ABSTRACT

Self-supervised pre-training of language models usually consists in predicting probability distributions over extensive token vocabularies. In this study, we propose an innovative method that shifts away from probability prediction and instead focuses on reconstructing input embeddings in a contrastive fashion via *Constrastive Weight Tying* (CWT). We apply this approach to pretrain Headless Language Models in both monolingual and multilingual contexts. Our method offers practical advantages, substantially reducing training computational requirements by up to 20 times, while simultaneously enhancing downstream performance and data efficiency. We observe a significant +1.6 GLUE score increase and a notable +2.7 LAMBADA accuracy improvement compared to classical LMs within similar compute budgets.

## 1 INTRODUCTION

Natural Language Processing (NLP) has seen tremendous progress in recent years thanks to the development of large-scale neural language models. These models have been shown to be effective in a wide range of NLP tasks such as text classification, question answering, and machine translation, either in fine-tuning, few-shot and zero-shot settings. These approaches usually involve a self-supervised pre-training step, based on tasks requiring predictions of contextual probability distributions over a large vocabulary of tokens.

However, the need for a language modeling projection head can be a limitation as it requires additional memory, slows down training and impedes scaling up to large token vocabularies. In this paper, we propose a novel pretraining approach called Headless Language Modeling, which removes the need to predict probability distributions and rather focuses on leveraging contrastive learning to reconstruct sequences of input embeddings. Instead of adding a projection head towards a high-dimensional vocabulary space in order to make a prediction about a given token, we teach those models to contrastively output static embeddings corresponding to this token. The static embeddings we use for this are the model's own input embeddings. Due to its resemblance with the well-established weight-tying trick (Press & Wolf, 2017; He et al., 2023), we call this pre-training technique *Contrastive Weight Tying* (CWT).

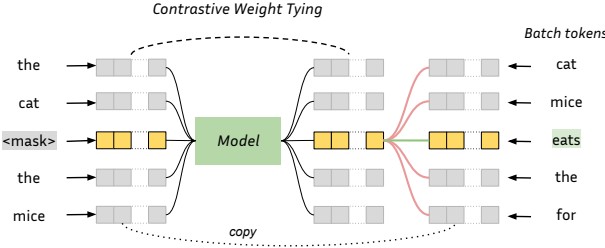

Figure 1: Masked Headless Language Modeling (HLM) using Contrastive Weight Tying. The CWT objective aims to contrastively predict masked input representations using in-batch negative examples.

We find that our approach outperforms usual language modeling counterparts in several aspects and by substantial margins. First, it drastically speeds up training by freeing up GPU memory and avoiding the costly language modeling projection, thus allowing up to $2\times$ acceleration of the training throughput, and up to $20\times$ less compute requirements to achieve similar performance. Moreover, given the same amount of training tokens, headless language models (HLMs) significantly outperform their classical counterparts on downstream tasks, as shown by a 2.7 gain in LAMBADA accuracy for our headless generative model. Finally, given similar compute budgets, HLMs bring substantial gains for NLU tasks, with our BERT reproduction scoring 1.6 points above its classical counterpart on the GLUE benchmark. We also show that headless models can benefit from larger token vocabularies at a much more reasonable cost than classical models.

In terms of implementation[1], our approach can be used as a drop-in replacement in usual pretraining codebases, as it only requires a change in the loss computation that can be applied to any kind of language model.

Overall, we make several contributions in this article:

- We introduce a pretraining objective that replaces cross-entropy, thus removing the need to project on the vocabulary high-dimensional space and instead learning to contrastively predict latent representations of tokens;
- Using this technique, we pretrain encoder and decoder models for English, and a multilingual encoder model;
- We show the various benefits of headless training in terms of data-efficiency, compute-efficiency, and performance;
- We explore the effects of micro-batch size and vocabulary size on downstream performance, and provide an ablation study of our contrastive objective.

## 2 RELATED WORK

**Efficient pre-training**  With the dawn of pretrained language models, such as BERT (Devlin et al., 2019), RoBERTa (Liu et al., 2019), GPT-2 (Radford et al., 2019) or T5 (Raffel et al., 2020), improving training efficiency has become an important stake in NLP. Subsequent works have focused on changing the training objectives to improve performance. ELECTRA (Clark et al., 2020b) uses Replaced Token Detection as the unsupervised training task, and substantially improves data-efficiency, compute-efficiency, and downstream performance. Their work has also been extended using energy-based models (Clark et al., 2020a) or disentangled weight sharing (He et al., 2020).

**Contrastive approaches in NLP**  The idea of relieving language models of the need to predict probabilities over the whole token vocabulary has been explored in the importance sampling literature (Bengio & Senecal, 2003; Mnih & Teh, 2012; Jean et al., 2015; Ma & Collins, 2018). These methods approximate the denominator of the softmax by using only a subset of the possible tokens. Those approaches usually rely on variants of the Noise-Contrastive Estimation objective (Gutmann & Hyvärinen, 2010) that use unique negative samples, contrary to our approach that samples representations uniformly from the batch. Kumar & Tsvetkov (2019) and Tokarchuk & Niculae (2022) use contrastive objectives based on cosine-similarity to match pre-trained static embeddings for Machine Translation. We instead use the model's input embeddings as trainable target representations.

**Contrastive self-supervised learning**  The Contrastive Predictive Coding loss (van den Oord et al., 2019) initiated the use of pretraining approaches based on a contrastive learning objective, an idea that has obtained success in many modalities over the years (Sermanet et al., 2018; Schneider et al., 2019; Baevski et al., 2020; Algayres et al., 2022). In NLP, contrastive learning has proven efficient in the training of sentence-level models (Gao et al., 2021; Yan et al., 2021; Klein & Nabi, 2023). Token-level approaches rely on contrastive auxiliary objectives that are added to the usual cross-entropy loss. SimCTG (Su et al., 2022a) introduces a token-level contrastive objective using in-batch output representations as negative samples, and adds this objective to a sentence-level contrastive loss and a regular causal LM loss. TaCL (Su et al., 2022b) relies on a similar technique for encoder models,

---

[1]Our pretraining and fine-tuning code is published in `https://github.com/NathanGodey/headless-lm`

where a teacher model is used to produce negative samples. ContraCLM (Jain et al., 2023) uses an auxiliary contrastive loss for code generation.

**Tokenization and frequency**  The importance of tokenization for language models has been discussed by several works (Rust et al., 2021; Zouhar et al., 2023). As discussed in Zouhar et al. (2023), tokenization choices impact token probability distributions both at contextual and general scales. It has been shown that skewed token distributions can impact the quality of representations (Gao et al., 2019a; Zhou et al., 2021; Puccetti et al., 2022; Yu et al., 2022). Removing the language modeling head could mitigate these issues. In the case of multilingual models, Liang et al. (2023) have shown that increasing the vocabulary size leads to better performance, at the cost of added time and memory complexity.

## 3   METHOD

### 3.1   CLASSICAL FRAMEWORK

We consider a batch $X = (x_{i,j})_{i \in [1,N], j \in [1,L]}$ of $N$ token sequences of length $L$. We also produce a slightly altered version of these sequences $\tilde{X} = (\tilde{x}_{i,j})_{i \in [1,N], j \in [1,\tilde{L}]}$, optionally using masking or random replacement for instance, as some pretraining objectives require. We introduce an embedding matrix $e_\theta \in \mathbb{R}^{V \times D}$ where $V$ is the token vocabulary size and $D$ is the hidden dimension, and a sequence-to-sequence model $T_\theta : \mathbb{R}^{N \times L \times D} \to \mathbb{R}^{N \times L \times D}$ both based on a set of parameters $\theta \in \mathbb{R}^P$.

A classical language modeling approach consists in selecting a subset of tokens $X_{\mathcal{S}} = (x_{i,j})_{i,j \in \mathcal{S}}$, and then estimating a probability distribution over the token vocabulary for these tokens from the $(\tilde{x}_{i,j})$ sequences, using $e_\theta$ and $T_\theta$. Learning occurs as $X_{\mathcal{S}}$ is partially altered in $(\tilde{x}_{i,j})$ (e.g. in Masked Language Modeling) or internally in $T_\theta$ (e.g. decoder models), and contextual information is essential for $e_\theta$ and $T_\theta$ to accurately estimate the tokens in $X_{\mathcal{S}}$.

A trick that has been used in many such approaches relies on using $e_\theta$'s transpose ($e_\theta^T$) as a projection from the output space of $T_\theta$ to $\mathbb{R}^V$. This approach, called weight tying, can be written for a given sequence at index $i \in [1,N]$ as:

$$\hat{p}_{i,j} = softmax\left(e_\theta^T \left(T_\theta(e_\theta(\tilde{x}_i))_j\right)\right)$$

where $\hat{p}_{i,j}$ is the estimated distribution for the $j$-th word of the sequence. Weight tying has been shown to improve performance while reducing the number of parameters (Clark et al., 2020b). Cross-entropy loss is then used as an objective function:

$$\mathcal{L}(\theta, X, \tilde{X}) = -\frac{1}{|\mathcal{S}|} \sum_{i,j \in \mathcal{S}} \mathbf{1}_{x_{i,j}} \cdot \log(\hat{p}_{i,j})$$

### 3.2   HEADLESS MODELING

While weight tying does not use additional parameters, the projection $e_\theta^T$ actually has a non-negligible computational cost, which increases as the token vocabulary grows. Like Gao et al. (2019a), we advocate that the weight tying approach tends to maximize the scalar product between the input embedding of the original token $e_\theta(x_{i,j})$ and the output representation at the same position $o_{i,j}^\theta = T_\theta(e_\theta(\tilde{x}_i))_j$, under the contrastive regularization of the softmax function.

Based on this understanding, we design an objective that directly optimizes this scalar product while not requiring the computation of the $e_\theta^T$ projection. As we do not use this projection, we cannot rely on softmax regularization anymore, and instead introduce a contrastive loss using the in-batch samples from $\mathcal{S}$ as negatives. All in all, our contrastive loss can be written as:

$$\mathcal{L}_c(\theta, X, \tilde{X}) = -\frac{1}{|\mathcal{S}|} \sum_{i,j \in \mathcal{S}} \frac{e^{o_{i,j}^\theta \cdot e_\theta(x_{i,j})}}{\sum_{k,l \in \mathcal{S}} e^{o_{i,j}^\theta \cdot e_\theta(x_{k,l})}}$$

We call this objective *Contrastive Weight Tying* (CWT), as weight sharing is not used *per se* but is set as a contrastive objective. Across the paper, we *do not combine* this loss function with the classical

cross-entropy objective as in Su et al. (2022a), and rather use it as the only pretraining objective. To the best of our knowledge, this work stands as the first attempt to pretrain language models in a self-supervised fashion using an explicit contrastive loss as the sole objective.

### 3.3 THE CASE OF DECODERS: CAUSAL FINE-TUNING

We can easily adapt the Causal Language Modeling (CLM) objective using the Contrastive Weight Tying approach. Negative samples correspond to every input embedding at a different position in the batch. However, the resulting model is not directly able to generate text, as it has no projection head towards $\mathbb{R}^V$. A way to retrieve language generation capacities is to use the input embedding matrix transpose $e_\theta^T$ as a projection head (Kumar & Tsvetkov, 2019; Tokarchuk & Niculae, 2022). Nevertheless, we observe that this approach yields poor performance (see Table 3). Instead, we fine-tune the headless model and a language modeling head initialized with $e_\theta^T$ using the predictive CLM objective on a small portion ($<$2%) of the pre-training dataset. This method allows recovering an effective language model.

### 3.4 THEORETICAL CONSIDERATIONS

In terms of time and memory complexity, Headless Language Models (HLMs) are more efficient than classical language models under usual conditions. If we focus on the computation of the loss *on a single device* from $|\mathcal{S}| = K$ output representations, a neural probabilistic LM requires $O(KDV)$ operations while our headless approach performs $O(K^2D)$ operations[2]. Hence, when $K < V$, which is very common for micro-batch sizes that fit on one device, our CWT loss is more computationally efficient than cross-entropy. With regard to memory requirements, our CWT loss is also more efficient than its classical counterpart. On the one hand, the cross-entropy loss with weight tying stores the outputs of the $e_\theta^T$ projection of dimension $K \times V$ in the forward pass. On the other hand, our CWT loss stores the scalar product matrix of dimension $K \times N$, which is again smaller when $K < V$.

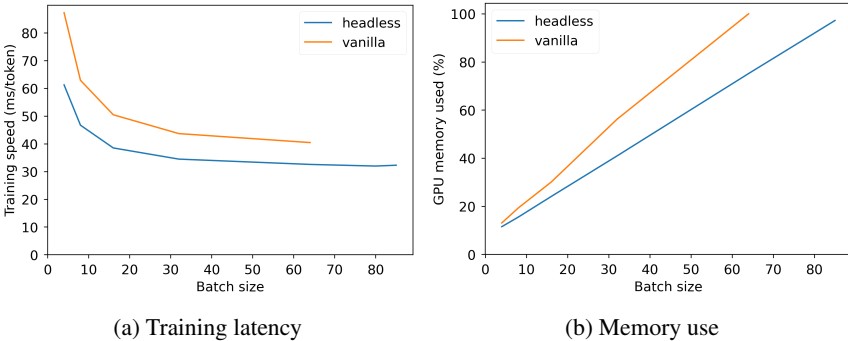

(a) Training latency          (b) Memory use

Figure 2: Comparison of time and memory complexities of a BERT-base model on a single RTX 8000 GPU.

In Figure 2, we provide a preliminary empirical analysis of the speed and memory improvements when training a BERT-base model using original hyperparameters, i.e. sequences of 512 tokens and 15% masking. We use HuggingFace's implementation for the Transformers blocks, and run experiments on a single RTX 8000 GPU. We observe that training latency is significantly reduced by roughly 25% for all batch sizes, and that the engine can handle a larger batch size due to the improvement in memory consumption.

## 4 EXPERIMENTS

We use the Contrastive Weight Tying objective for medium-scale pre-training experiments in different contexts. We focus on monolingual encoder and decoder architectures, but we also train one

---

[2]One could extend our CWT loss by picking a separate set $\mathcal{S}_N$ of negative samples. This allows to tune the number of negative samples, which is important in Contrastive Learning. However, for the sake of simplicity, and to avoid extensive hyperparameter tuning, we set $\mathcal{S}_N = \mathcal{S}$.

| MLM type | Tokens (B) | GPU hours | MRPC | COLA | STS-B | SST2 | QNLI | QQP | MNLI | Avg. |
|---|---|---|---|---|---|---|---|---|---|---|
| Vanilla | 4.1 | 989 | 85.87 | 54.66 | 83.7 | 92.45 | 88.38 | 89.57 | 82.4 | 82.43 (±0.12) |
| Headless | 4.1 | 444 | 85.31 | 58.35 | 84.54 | **93.23** | 89.49 | 89.62 | 82.54 | 83.29 (±0.15) |
| Headless | 8.2 | 888 | **86.89** | **60.72** | **85.98** | 92.56 | **89.75** | **89.81** | **82.87** | **84.08** (±0.14) |

Table 1: Results of Masked Language Models (MLMs) on the dev sets of the GLUE benchmark. Best results are **bold** and second best are underlined. We report Matthews' correlation for COLA, Spearman correlation for STS-B, and accuracy elsewhere. MNLI validation datasets are concatenated. All scores are averaged over 3 different seeds.

| MLM type | BoolQ | CB | COPA | WiC | Avg. |
|---|---|---|---|---|---|
| Vanilla | 68.8 | **77.8** | 60.2 | 64.9 | 67.9 (±0.4) |
| Headless | **69.8** | 74.7 | **62.7** | **67.2** | **68.6** (±0.6) |

Table 2: Results of Masked Language Models (MLMs) on the dev sets of datasets from the Super-GLUE benchmark. We report accuracy for all tasks. Scores are averaged over 10 fine-tuning runs.

multilingual encoder as we believe the uniformity brought by our contrastive objective may improve cross-lingual alignment. We compare our HLMs with classical language models that we pretrain on the same data with roughly similar compute budgets.

## 4.1 HEADLESS MONOLINGUAL ENCODER

We pretrain BERT-base architectures (110M parameters) for English on the OpenWebText2 dataset extracted from The Pile (Gao et al., 2020). We use the tokenizer from the Pythia suite (Biderman et al., 2023), which was trained on The Pile and uses a 50k tokens vocabulary. We mostly use hyperparameters from BERT (Devlin et al., 2019), although we remove the NSP objective as in RoBERTa (Liu et al., 2019). For the sake of simplicity, we use a sequence length of 128 for the whole training. We give a detailed overview of the hyperparameters in Appendix D.1.

We pretrain all models using 8 A100 GPUs, with a budget of roughly 1,000 hours each. To optimize training, we use memory-efficient self-attention as implemented in xFormers (Lefaudeux et al., 2022) for all experiments. For the vanilla MLM, we set a micro-batch size of 32 for each A100 GPU, then accumulate to the original 256 batch size at optimization level, and train on 1 million batches. For our headless approach, we observed that we could remain within compute budget when using a micro-batch size of 64. Hence, we use an effective batch size of 512 for the headless MLM (HMLM). Although the HMLM uses more pretraining sequences, it does not gain additional information compared to the vanilla MLM as both models perform several epochs on the OpenWebText2 dataset.

We evaluate on the GLUE benchmark, where we exclude the RTE dataset due to high standard deviations in the obtained scores. We fine-tune our models for 10 epochs on every dataset, and compute validation metrics once every fine-tuning epoch. We use the AdamW optimizer with a learning rate of $10^{-5}$, a weight decay of 0.01 and a balanced cross-entropy loss objective. See Appendix E for more details.

In Table 1, we compare our headless MLM with the classical MLM on the GLUE benchmark. To ensure fair comparison, we display evaluations at similar amounts of tokens seen during pre-training, and at similar training durations on the same hardware. In both cases, the headless MLM outperforms the vanilla MLM by significant margins, showing that our CWT loss is both more data-efficient and compute-efficient in this setup. We extend this analysis at various intervals along pretraining, and plot results in Figure 3. It shows that the headless MLM outperforms the downstream performance of its vanilla counterpart after using 25% of its training compute. We notice that the performance gap is near constant across pretraining steps.

## 4.2 HEADLESS MONOLINGUAL DECODER

We pretrain Pythia-70M architectures for English, sticking to the Pythia procedure (Biderman et al., 2023) as much as possible. We use OpenWebText2 as a pretraining dataset. We train on 143,000

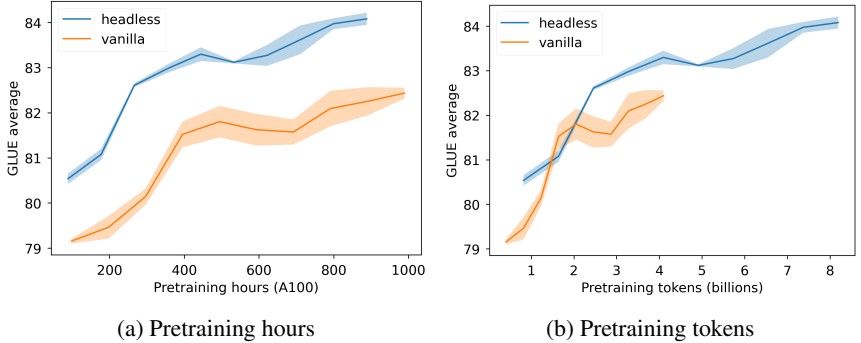

(a) Pretraining hours        (b) Pretraining tokens

Figure 3: Comparison of GLUE average scores along pretraining.

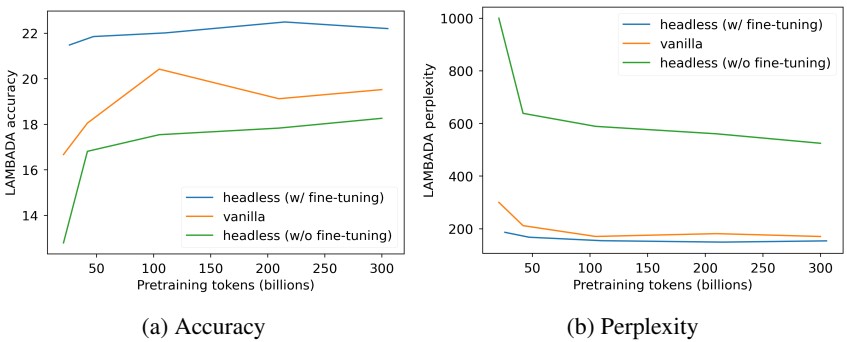

(a) Accuracy        (b) Perplexity

Figure 4: Comparison of LAMBADA metrics along pretraining. We display results for vanilla causal language modeling and headless models before and after causal LM fine-tuning. The pretraining token count for the fine-tuned HLM takes fine-tuning tokens into account.

batches of 1,024 sequences of length 2,048 split over 16 V100 GPUs. We use exactly the same hyperparameters as in the Pythia suite. The micro-batch size is set to 32 in both cases.

As mentioned in Subsection 3.3, we fine-tune our headless models for CLM with an LM head initialized with $e_\theta^T$ for 10000 steps using an effective batch size of 256 ($4\times$ smaller that during pretraining), a learning rate of $10^{-4}$, and a constant learning rate schedule with 2000 linear warm-up steps. All other hyperparameters are kept similar to pretraining. We evaluate our models on the LAMBADA dataset and report accuracy and perplexity for zero-shot generation in Figure 4.

We find that the HLM fine-tuned for predictive language modeling outperforms the vanilla model by a significant margin along training. We report language generation results in Table 3. We observe that despite having a higher validation perplexity even after fine-tuning, the HLM is improving the zero-shot perplexity on the LAMBADA dataset.

We also study the zero-shot performance of the causal models on datasets taken from the LM Evaluation Harness. At this model scale, many tasks are not relevant and thus discarded, as the results do not always significantly outperform a random baseline. We also discarded tasks where the

| LM type | Validation | LAMBADA | |
|---|---|---|---|
| | Ppl. | Ppl. | Acc. |
| Vanilla | **3.143** | 170.23 | 19.52 |
| Headless | - | 524.44 | 18.26 |
| Headless + FT | 3.283 | **153.5** | **22.2** |

Table 3: Results of the causal language models on the validation set after training, and on the LAMBADA dataset.

| LM type | GPU hours | BoolQ | CrowS-Pairs ↓ | RACE | SciQ | PubMedQA | QASPER |
|---|---|---|---|---|---|---|---|
| Vanilla | 1712 (-) | 47.8 (±0.9) | 57.3 (±1.2) | 23.7 (±1.3) | **66.4** (±1.5) | 43.8 (±1.6) | 41.9 (±4.8) |
| HLM + FT | 1052 (61%) | **53.0**[†] (±0.9) | **56.0** (±1.2) | **26.0** (±1.4) | 64.5 (±1.5) | **47.5**[†] (±1.6) | **66.0**[†] (±3.1) |

Table 4: Zero-shot evaluation of monolingual causal language models on datasets from the LM Evaluation Harness. We report the stereotype percentage for CrowS-Pairs and accuracy elsewhere. [†]: best scores that are significantly better than the second best score according to a one-tailed t-test with power 0.95.

sample size was below 1000 or where comparison was not meaningful due to low performance gaps compared to the variance level. Hence, only a subset of the tasks is shown in Table 4.

In Table 4, we find that the fine-tuned HLM outperforms the vanilla causal model by significant margins on BoolQ (Clark et al., 2019), PubMedQA (Jin et al., 2019) and QASPER (Dasigi et al., 2021). Although we observe less statistically significant gaps for the other datasets, we still note that our HLM performs at least comparably to the vanilla baseline. We also note that the HLM seems slightly less prone to stereotypes as measured by the CrowS-Pairs benchmark (Nangia et al., 2020).

Overall, using the Contrastive Weight Tying loss in the context of causal LM allows obtaining models on par with vanilla counterparts at a lower compute cost. We notice that the resulting models can get surprisingly good results in challenging datasets, hence showing language understanding capabilities, while being outclassed in language generation benchmarks (before predictive fine-tuning). We believe that this study shows that language generation needs to be considered as a *downstream task* for HLMs, as they are designed to generate representations instead of words.

## 5 MULTILINGUAL ENCODER

In this section, we pretrain small multilingual MLMs and evaluate their performance on the XNLI dataset (Conneau et al., 2018). Due to compute limitations, we consider architectures similar to the distilled multilingual BERT[3] trained by Sanh et al. (2019). This model has 137M parameters, and uses a vocabulary of 119k tokens. As in Subsection 4.1, we train a vanilla MLM and a headless counterpart. However, we share training hyperparameters such as batch size and total number of steps between both models, without compute considerations. For both experiments, we pretrain our models on 400k batches of 64 sequences of 128 tokens taken from the multilingual Wikipedia dataset using a single RTX8000 GPU. We select 90 million entries from 10 languages (Arabic, German, English, Spanish, French, Hindi, Italian, Japanese, Korean, and Chinese). Training hyperparameters can be found in Appendix D.3.

Models are then fine-tuned on the XNLI dataset, for both cross-lingual zero-shot transfer from English and target language fine-tuning. Fine-tuning hyperparameters can be found in Appendix E.4.

| MLM type | ar | de | en | es | fr | hi | zh | Avg. |
|---|---|---|---|---|---|---|---|---|
| *Fine-tuned on English only* | | | | | | | | |
| Vanilla | 46.83 | 56.71 | 71.66 | 59.93 | 58.34 | 43.16 | 50.99 | 55.37 (±0.11) |
| Headless | **48.06** | **57.32** | **74.03** | **62.72** | **62** | **45.25** | **52.15** | **57.36** (±0.2) |
| *Fine-tuned on target language* | | | | | | | | |
| Vanilla | 51.32 | 64.09 | 70.4 | 66.98 | 65.88 | 55.95 | 64.63 | 62.87 (±0.2) |
| Headless | **54.25** | **66.95** | **73.96** | **69.14** | **67.22** | **60.04** | **67.22** | **65.54** (±0.22) |

Table 5: Evaluation of multilingual models on the XNLI benchmark. We report dev accuracy, averaged over 3 runs.

We display final results in Figure 5. We find that the headless approach leads to significantly better performance for every language in both cross-lingual transfer and language-specific fine-tuning. In average, the headless MLM outperforms its vanilla counterpart by 2 accuracy points in the cross-lingual scenario, and by 2.7 points in the language-specific fine-tuning experiments.

---

[3]Available at `https://huggingface.co/distilbert-base-multilingual-cased`

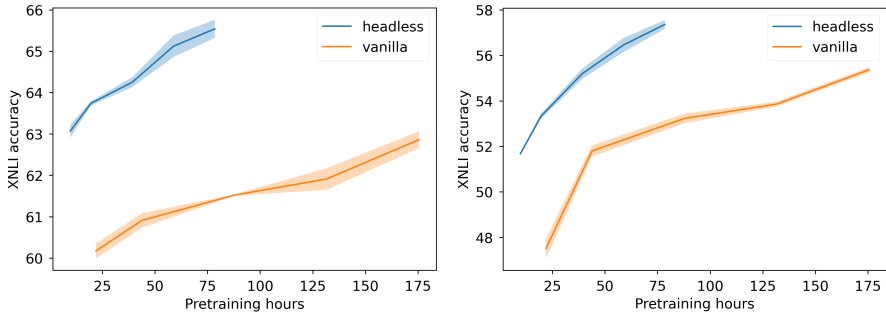

(a) Translate-Train: target language fine-tuning    (b) Translate-Test: English fine-tuning

Figure 5: Comparison of XNLI average scores along pretraining for different setups. Models are fine-tuned/evaluated in Arabic, German, English, Spanish, French, Hindi and Chinese.

In Figure 5, we evaluate the models at intermediate pretraining checkpoints and plot the XNLI average score as a function of used GPU hours. We observe that our HLM finishes training within 45% of the time required by the vanilla mode, and that its performance level outperforms the fully trained vanilla model after only using 5% as much compute in Figure 5a, and 22% in Figure 5b.

## 6    DISCUSSION

**Token vocabulary**    Training language models without output vocabulary projection makes using large vocabularies more affordable in terms of compute. As a matter of fact, the time complexity of HLMs during training is theoretically constant as we increase the vocabulary size. With input embedding lookup tables that do not require fully loading the $e_\theta$ weights, the memory complexity can also be kept constant with respect to the size of the vocabulary. This property could be useful for multilingual models relying on considerable vocabulary sizes, such as XLM-V (Liang et al., 2023).

To verify this hypothesis, we pretrain models for different vocabulary sizes using the BERT-Small architecture from Turc et al. (2019) and the CC-News dataset (Hamborg et al., 2017). Hyperparameter details can be found in Appendix D.4. For each vocabulary size, we train a BPE tokenizer similar to the BERT tokenizer, and pretrain a vanilla MLM and a headless MLM. We then compare average GLUE results, excluding RTE, MRPC and COLA, due to high variance at that model scale.

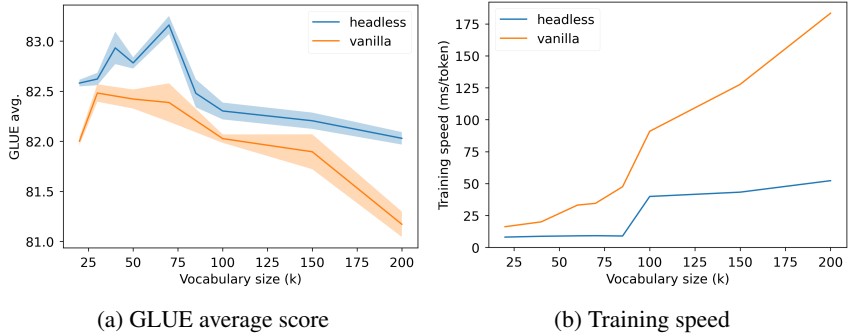

(a) GLUE average score    (b) Training speed

Figure 6: Comparison of downstream performance and training speed for small models trained using different token vocabulary sizes.

Figure 6 shows that HLMs can actually benefit from larger token vocabularies up to a certain extent, and that they outperform their vanilla counterparts for every vocabulary size. Figure 6b demonstrates that increasing the vocabulary size comes at almost no decrease in training speed for the HLMs, contrary to vanilla MLMs. However, we observe a sudden throughput increase between 85k and 100k tokens vocabularies for both vanilla and headless models, which we attribute to a different handling of GPU memory and operations as the models get bigger.

**Batch size**    As discussed in Subsection 3.4, the micro-batch size used to compute the CWT loss is important as it impacts the training complexity by increasing the number of negative samples. Recent work on Contrastive Learning shows that there usually exists an optimal number of negative samples in terms of model performance (Awasthi et al., 2022; Ash et al., 2022). As a consequence, increasing the batch size when using CWT may not always be beneficial.

To study the impact of batch size on downstream performance, we pretrain small decoder models using different batch sizes. Our models are inspired from the smallest architecture of GPT2 (Radford et al., 2019) where many hyperparameters are divided by 4. More details about the pretraining procedure of these models can be found in Appendix D.5. HLMs are fine-tuned similarly to Subsection 4.2.

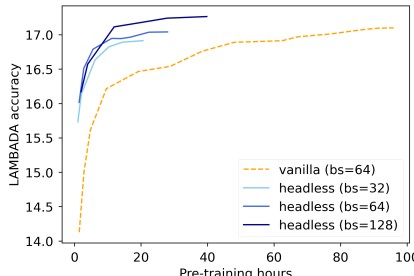

Figure 7: LAMBADA accuracy along pretraining for different batch sizes.

In Figure 7, we observe that increasing batch size leads to better performance for our HLMs. While smaller batch sizes train even faster, the headless model with the greatest batch size (128) is the only one that is able to significantly outperform its vanilla counterpart at the end of training.

**Ablation study**    In Table 6, we conduct an ablation study by training small models using the hyperparameters described in Appendix D.4 for different objectives. We observe that adding Cross-Entropy to CWT leads to slightly worse performance, at the cost of reduced throughput. We also notice that using a contrastive objective without using input embeddings as targets decreases performance, despite adding parameters during training. This shows the relevance of our weight tying approach.

| Objective | Parameters | Throughput ↑ | GLUE avg. |
|---|---|---|---|
| Cross-Entropy | x1 | x1 | 82.45 |
| Cross-Entropy + CWT | x1 | x0.87 | 82.93 |
| NCE (wo/ WT) | x1.57 | **x2.47** | 82.91 |
| CWT | x1 | **x2.13** | **83.37** |

Table 6: Ablation study using variants of the CWT objective. In CWT + Cross-Entropy, we add the objectives without specific weights. In NCE (wo/ WT), we adapt our CWT objective with an additional static embedding matrix instead of the model's input embeddings, which resembles Ma & Collins (2018).

## CONCLUSION

In this paper, we present a new pretraining approach called headless language modeling, that removes the need to predict probability distributions over token vocabulary spaces and instead focuses on learning to reconstruct representations in a contrastive fashion. Our method only relies on changing the objective function, allowing for straightforward adaptations of classical language modeling pretraining objectives.

Using our contrastive objective, we pretrain headless monolingual and multilingual encoders, and a headless monolingual decoder. We demonstrate that headless pretraining is significantly more compute-efficient, data-efficient, and performant than classical predictive methods.

A major advantage of our approach is that it enables the use of very large token vocabularies at virtually no increased cost. We believe that this paper paves the way for the exploration of contrastive techniques as a replacement of cross-entropy based pretraining objectives for NLP.

ACKNOWLEDGEMENTS

We thank our colleagues Arij Riabi and Roman Castagné for their advice and for the helpful discussions. We are grateful to Robin Algayres for his enlightening question *"But what is the difference with softmax?"*, in the hope that this paper is a satisfying answer.

This work was funded by the last author's chair in the PRAIRIE institute, itself funded by the French national agency ANR as part of the "Investissements d'avenir" programme under the reference ANR-19-P3IA-0001.

This work was granted access to the HPC resources of IDRIS under the allocation 2023-AD011013680R1 made by GENCI.

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

## A  MODELING CONSIDERATIONS

From a linguistic point of view, we hypothesize that an important difference between our approach and classical predictive modeling is the fact that *headless modeling mostly pushes for discrimination between co-occurring tokens*, instead of imposing a contextual hierarchy over the whole vocabulary. For instance, in the case of synonyms A and B, each occurrence of A (or B) is pushing the input representations of A and B apart for predictive modeling, due to weight tying. For headless modeling, an occurrence of A will only push the representations apart if B appears in the same batch. Hence, the CWT objective could let models identify A and B as synonyms more easily. This argument is already mentioned in Jean et al. (2015).

To provide empirical evidence of this behavior, we study the representation similarity for pairs of synonyms for classical and headless models. We use WordNet (Fellbaum, 1998) to extract synonym pairs and we then compute the cosine-similarity between the input embeddings corresponding to the two synonyms. Resulting cosine-similarity distributions are displayed in Figure 8.

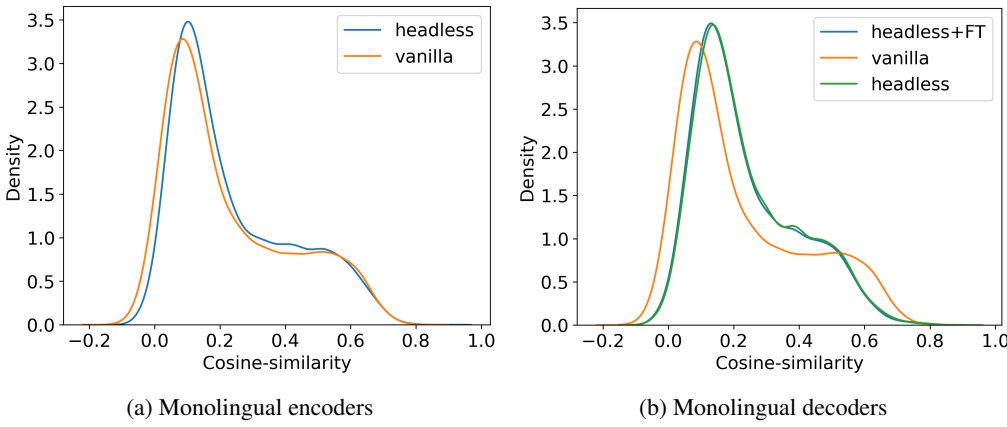

(a) Monolingual encoders       (b) Monolingual decoders

Figure 8: Cosine-similarity distributions for pairs of WordNet synonyms.

In Figure 8, we observe that HLMs tend to generally represent synonyms in a more similar way than vanilla LMs, as cosine-similarity distributions slightly drift towards higher values. In average,

cosine-similarity between synonyms is 1.4 points higher for the encoder and roughly 7 points higher for both the original HLM decoder and its fine-tuned version.

However, we do not observe a radical difference between HLMs and classical LMs in this analysis of the input representations. A more thorough analysis of the latent spaces of both types of models could be relevant. For instance, comparing contextual representations of similar words across examples could help clarify this matter. We leave such analyses for future work.

Another advantage of pushing discrimination between co-occurring tokens only may be an improved feedback quality, as we expect distinguishing between co-occurring tokens to be more linguistically relevant than distinguishing between all tokens.

Finally, we believe that our method avoids the issue of cross-entropy regarding rare and unused tokens. Gao et al. (2019b) prove that cross-entropy pushes the representations of rare and unused tokens in a shared direction, thus distorting the resulting embedding space. The CWT objective only updates these embeddings when they appear in the negative samples, which should result in more meaningful representations.

## B    LIMITATIONS

One key limitation of this paper is the scale of the used architectures. In recent months, the dawn of Large Language Models using billions of parameters reshaped the language modeling paradigm. The research process that led to this paper is empirical and required extensive experimentation that could not be done at large scale in our academic compute budget. We believe that the results presented in this paper are still sufficiently promising to be communicated and useful to the community. We leave the scaling of these techniques to future work.

It could be opposed to this paper that as architectures grow in size, the proportion of compute that is associated with the output vocabulary projection shrinks. While we acknowledge that this effect may reduce the advantage of HLMs in terms of training throughput, our experiments show that HLMs are more performant for a given number of pretraining steps.

We chose not to compare with other efficient encoder architectures such as ELECTRA or DeBERTa in this paper. We also chose not to apply our method to encoder-decoder architectures, or to subtle masking methods such as SpanBERT (Joshi et al., 2020). As a matter of fact, we argue that our work could be combined to these methods, and we thus believe that comparison is not relevant as these works are orthogonal to ours. We leave the intersection of these approaches for future work.

Finally, we decided to pick English for all monolingual experiments. Different behaviors could be observed for other languages, although our multilingual experiments gave no sign of such discrepancies.

## C    ETHICS STATEMENT

To the best of our knowledge, this paper does not raise any specific ethical concern that is not already inherent to the open-data pre-training paradigm. Our results on the CrowS-Pairs dataset indicate that headless language modeling may mitigate some of the biases that are measured in this task. Due to considerations that are discussed in Zhou et al. (2021), and for reasons evoked in Section 6, we believe that alternatives to cross-entropy as an objective for language modeling could mitigate some of the biases that are observed in LLMs, and hope that our work can pave the way for such alternatives.

# D   PRETRAINING HYPERPARAMETERS

## D.1   MONOLINGUAL ENCODERS

| Dataset | OpenWebText2 |
|---|---|
| Architecture | bert-base-uncased |
| Tokenizer | pythia-70m-deduped |
| Optimizer | AdamW |
| Learning rate | 1e-4 |
| Precision | 16 |
| Weight decay | 0.01 |
| Gradient clipping | 1 |
| Device batch size | 32 / 64 |
| Batch size | 256 / 512 |
| Sequence length | 128 |
| LR schedule | Triangular |
| Warmup steps | 10000 |
| Nb. steps | 1000000 |

Table 7: Pre-training hyperparameters used for the monolingual encoders. When they differ between vanilla and headless models, we provide separate values formatted as (vanilla / headless). Model names written as model-name refer to their HuggingFace release.

## D.2   MONOLINGUAL DECODERS

| Dataset | OpenWebText2 |
|---|---|
| Architecture | pythia-70m-deduped |
| Tokenizer | pythia-70m-deduped |
| Optimizer | AdamW |
| Adam $\epsilon$ | 1e-8 |
| Adam $(\beta_1, \beta_2)$ | (0.9, 0.95) |
| Learning rate | 1e-3 |
| Precision | 16 |
| Weight decay | 0.1 |
| Gradient clipping | 1 |
| Device batch size | 8 / 8 |
| Batch size | 1024 / 1024 |
| Sequence length | 2048 |
| LR schedule | Cosine |
| Warmup steps | 1430 |
| Nb. steps | 143000 |

Table 8: Pre-training hyperparameters used for the monolingual encoders. When they differ between vanilla and headless models, we provide separate values formatted as (vanilla / headless).

## D.3 MULTILINGUAL ENCODERS

| Dataset | Wikipedia (multilingual) |
|---|---|
| Architecture | `distilbert-base-multilingual-cased` |
| Tokenizer | `distilbert-base-multilingual-cased` |
| Optimizer | AdamW |
| Learning rate | 2e-4 |
| Precision | 16 |
| Weight decay | 0.01 |
| Gradient clipping | 1 |
| Device batch size | 64 |
| Batch size | 64 |
| Sequence length | 128 |
| LR schedule | Triangular |
| Warmup steps | 10000 |
| Nb. steps | 400000 |

Table 9: Pre-training hyperparameters used for the multilingual encoders.

## D.4 SMALL MONOLINGUAL ENCODERS

| Dataset | CC-News |
|---|---|
| Architecture | `google/bert_uncased_L-4_H-512_A-8` |
| Tokenizer | `google/bert_uncased_L-4_H-512_A-8` |
| Optimizer | AdamW |
| Learning rate | 2e-4 |
| Precision | 16 |
| Weight decay | 0.01 |
| Gradient clipping | 1 |
| Device batch size | 64 |
| Batch size | 64 |
| Sequence length | 128 |
| LR schedule | Triangular |
| Warmup steps | 10000 |
| Nb. steps | 400000 |

Table 10: Pre-training hyperparameters used for the small monolingual encoders used in Figure 6.

## D.5 SMALL MONOLINGUAL DECODERS

| Dataset | CC-News |
|---|---|
| Architecture | gpt2 |
| Hidden size | 192 |
| Number heads | 3 |
| Number layers | 3 |
| Tokenizer | gpt2 |
| Optimizer | AdamW |
| Learning rate | 2.5e-4 |
| Precision | 16 |
| Weight decay | 0.01 |
| Gradient clipping | 1 |
| Sequence length | 128 |
| LR schedule | Cosine |
| Warmup steps | 2000 |
| Nb. steps | 1000000 |

Table 11: Pre-training hyperparameters used for the small monolingual decoders used in Figure 7. These models rely on the GPT-2 architecture with a few changes. These changes scale down the model size to 11M parameters.

## E FINETUNING HYPERPARAMETERS

### E.1 BALANCED CROSS-ENTROPY

We have noticed that using balanced cross-entropy loss for fine-tuning could further improve the performance of all our monolingual encoders, and increase the gap between headless models and their vanilla counterparts. We also noticed empirically that it helped stabilize results for smaller datasets such as MRPC and COLA.

Let's consider a classification problem where the class distribution is described by frequencies $(w_c)_{c \in [1,C]}$. We can group the cross entropy loss $\mathcal{L}_{ce}$ as such:

$$\mathcal{L}_{ce}(X, Y) = \sum_{c=1}^{C} \mathcal{L}_c(X, Y)$$

where

$$\mathcal{L}_c(X, Y) = \sum_{i=1}^{N} \mathbf{1}_{y_i=c} \cdot \mathcal{L}_{ce}(x_i, y_i)$$

Using this notation, the *balanced cross-entropy loss* can be defined as:

$$\mathcal{L}_{bce}(X, Y) = \sum_{c=1}^{C} \frac{\mathcal{L}_c(X, Y)}{w_c}$$

In practice, we approximate the $(w_c)$ using the batch labels. The purpose of the balanced cross-entropy loss is to mitigate general and in-batch class imbalance.

We reproduce fine-tuning experiments with the more usual categorical cross-entropy loss only, and using moderately optimized hyperparameters for this loss (see Table 12).

| Optimizer | AdamW |
|---|---|
| Learning rate | 5e-6 |
| Weight decay | 0.01 |
| Batch size | 32 |
| LR schedule | Constant |
| Linear warm-up | 10% |
| Epochs | 10 |

Table 12: Fine-tuning hyperparameters for monolingual encoder models trained with regular cross-entropy on the GLUE benchmark.

| MLM type | MRPC | COLA | STS-B | SST2 | QNLI | QQP | MNLI | Avg. |
|---|---|---|---|---|---|---|---|---|
| Vanilla | **86.27** | 49.33 | 82.06 | 92.37 | 88.62 | 89.49 | 82.35 | 81.5 (±0.14) |
| Headless | 85.8 | **56** | **84.85** | **93.23** | **89.67** | **89.77** | **83.05** | **83.19** (±0.09) |

Table 13: Results of Masked Language Models (MLMs) on the dev sets of the GLUE benchmark for the regular cross-entropy loss. Results are averaged over 3 runs.

## E.2 MONOLINGUAL ENCODERS

| Optimizer | AdamW |
|---|---|
| Learning rate | 1e-5 |
| Cross-entropy | Balanced |
| Weight decay | 0 |
| Batch size | 32 |
| LR schedule | Constant |
| Linear warm-up | 10% |
| Epochs | 10 |

Table 14: Fine-tuning hyperparameters for monolingual encoder models trained with balanced cross-entropy on the GLUE benchmark.

## E.3 MONOLINGUAL DECODERS

| Dataset | OpenWebText2 |
|---|---|
| Optimizer | AdamW |
| Learning rate | 1e-5 |
| Cross-entropy | Regular |
| Weight decay | 0 |
| Batch size | 256 |
| LR schedule | Constant |
| Linear warm-up | 2000 |
| Nb. steps | 10000 |

Table 15: Fine-tuning hyperparameters for the headless monolingual decoder model using the causal language modeling objective.

### E.4 MULTILINGUAL ENCODERS

| Optimizer | AdamW |
|---|---|
| Learning rate | 2e-5 |
| Cross-entropy | Regular |
| Weight decay | 0 |
| Batch size | 128 |
| LR schedule | Constant |
| Linear warm-up | 10% |

Table 16: Fine-tuning hyperparameters for the multilingual encoder models in Translate-Train and Translate-Test scenarios.

## F REPRESENTING SYNONYMS

In this section,

## G IMPLEMENTATION

```python
def cwt_loss(input_embs, target_embs):
    # input_embs: nb_embs x hidden_dim
    # target_embs: nb_embs x hidden_dim

    exp_cosine_sim = torch.exp(torch.mm(input_embs, target_embs.T))
    self_dist = exp_cosine_sim.diagonal()
    neg_dist = exp_cosine_sim.sum(-1)

    return - (self_dist/(neg_dist + 1e-9)).log().mean()
```

Figure 9: PyTorch implementation of the Contrastive Weight Tying loss.

The figures were generated using the Carbon tool (`https://carbon.now.sh/`).

```python
def compute_loss(lm_model, input_batch):
    # input_batch: batch_size x seq_length (LongTensor)

    labels = input_batch[..., 1:]

    # Get model output
    lm_result = lm_model(input_batch, output_hidden_states=True)
    last_hidden_state = lm_result.hidden_states[-1][:, :-1]

    # Get input embeddings
    emb_mapping = lm_model.get_input_embeddings()
    target_input_embeddings = emb_mapping(labels)

    # Compute CWT loss
    batch_loss = cwt_loss(
        emb_prediction.flatten(0, 1),
        target_input_embeddings.flatten(0, 1)
    )

    return batch_loss
```

Figure 10: PyTorch implementation of the computation of the training loss for headless causal LMs. The implementation of the MLM equivalent is straightforward.

