# OpenReview forum: "Headless Language Models: Learning without Predicting with Contrastive Weight Tying"
_ICLR.cc/2024/Conference — ICLR 2024 poster_

### Official Review · Reviewer_QSPQ · 2023-10-31

**Soundness:** 4 excellent
**Presentation:** 3 good
**Contribution:** 3 good
**Rating:** 8
**Confidence:** 3

**Summary:**

The paper proposes a simple method for pre-training LMs with contrastive loss. Instead of evaluating logits for all vocabulary, only embeddings from the batch are used. Such an approach allowed us to pre-train LMs more effectively by reducing memory constraints. Surprisingly, this approach also performed well for autoregressive language modeling. For both autoregressive and masked language modeling, the proposed method showed improved results on downstream tasks, as well as improved training efficiency with respect to training hours.

**Strengths:**

The proposed method is straightforward yet shows promising results.
Performance on down-stream tasks is marginally improved with respect to vanilla models
Pre-training is more efficient with respect to training time and the total number of tokens
The paper is easy to follow, and the motivation is clear.

**Weaknesses:**

I would like to see more insights from the paper. It shows that HLMs provide better efficiency and down-stream performance, while such results are slightly contr-intuitive. More analysis could make the paper better. I.e., it would be beneficial to answer the question, "what makes HLMs special, and why it outperforms vanilla LMs?".

**Questions:**

Please, refer to the Weaknesses section

---

> ### Author Response · Authors · 2023-11-13
> **Rebuttal**
>
> We warmly thank the reviewer for its enthusiasm with regard to our work. We addressed the main question of the reviewer in **Appendix A** and in the General rebuttal.

---

### Official Review · Reviewer_CHGU · 2023-10-31

**Soundness:** 3 good
**Presentation:** 1 poor
**Contribution:** 2 fair
**Rating:** 6
**Confidence:** 5

**Summary:**

This paper proposes to replace the cross-entropy minimization by a new pre-training objective in language models: Contrastive Weight Tying (CWT). It consists in contrastively predicting the representation of the missing word. Instead of projecting the representation output by the model into the vocabulary, and using a softmax, the goal is to predict directly this static embedding of the reference word, which the model is trained for by (1) using a contrastive objective with the rest of the batch as negative examples (2) using input static embeddings as reference output embeddings, which is traditionally referred as *weight tying*. After a presentation of previous work, the paper introduces the method, consisting in weight tying, the contrastive objective with no projection, and how to adapt it for text generation, plus computational requirements. Experiments are performed on a Monolingual Encoder (GLUE), Monolingual Decoder (LAMBADA/LM Evaluation Harness), and Multilingual Encoder (XNLI Benchmark), with CWT showing improvements on all settings compared to the model trained classically. Lastly, the paper presents experiments on vocabulary size, as expanding the vocabulary is less costly and may provides some advantages with CWT.

**Strengths:**

- This paper presents a simple and well motivated method for pre-training a language model, CWT.
- The paper implements various experiments showing that CWT consistently improves performance, training efficiency, or both, over the classical version of the model.

**Weaknesses:**

The main issue I found with this paper is a huge lack of bibliographic work. It mainly keeps the paper from proper contextualization but I believe a proper related work would have greatly helped in making the experiments less shallow.
- The lack of reference to relevant previous work keeps the paper from being well-positioned within the literature, especially when it comes to comparison with the numerous pre-existing similar methods,
- It also keeps it from investigating relevant experimental settings - for example, comparison with similar objectives and ablation studies which could shed some light on what makes CWT work better than some previous methods.

**Questions:**

- On the missing related work: I will detail what I estimate to be essential. I believe you could find most of these references (and many other) in papers you have yourself cited (the papers describing ELECTRA and ELECTRIC, if only).
    - You write "This work stands as the first attempt to train language models using an explicit contrastive loss".  The first that I am aware of would be [1], but there are many more recent references, especially those making use of Noise Contrastive Estimation; with [2] which, while binary in nature, has been adapted into a loss resembling your own in [3], though I believe the first use of a direct softmax-like objective with negative examples coming from the batch is from [4].
    - Though I believe input/output embeddings weight sharing was a common approach that was very certainly implemented in most early languages models prior to 2017, your work combines this approach with a contrastive objective in order to get rid of the output vocabulary, which is a very nice idea. On that matter, I will mainly cite [5], a dedicated loss aiming at predicting static embeddings too; another paper [6] studies how to choose the target embeddings (which you use weight tying for).
    - Language modelling without an output vocabulary (contrastive objective or not) has been explored quite a lot in the past, with the specific problem you address in subsection 3.3 with fine-tuning having had many proposed solutions. I believe a nearest neighbour approach has been used in [5] and [6], but [7] proposes an energy-based model to repurpose a well-trained encoder (outputting *embeddings*) into a generative model.
    - I believe you will find in this previous work many insights which will help you improve the design of your experiments and directions for improvement.

- On experiments:
    - Do you have any hypothesis regarding the better performance of CWT than the vanilla model *with the same number of training token* on most GLUE tasks ?
    - In fine-tuning a causal LM for text generation, is the supplementary amount of compute received by the fine-tuned model compared to the vanilla model significant ?




- [1] Quick Training of Probabilistic Neural Nets by Importance Sampling (Bengio and Sénacal, 2003)
- [2] A fast and simple algorithm for training neural probabilistic language models (Mnih and Teh, 2013)
- [3] Noise Contrastive Estimation and Negative Sampling for Conditional Models: Consistency and Statistical Efficiency (Ma and Collins, 2018)
- [4] On Using Very Large Target Vocabulary for Neural Machine Translation (Jean et al, 2014)
- [5] Von Mises-Fisher Loss for Training Sequence to Sequence Models with Continuous Outputs (Kumar and Tsekov, 2019)
- [6] On Target Representation in Continuous-output Neural Machine Translation (Tokarchuk and Niculae, 2022).
- [7] Residual Energy-Based Models for Text Generation (Deng et al, 2019)

---

> ### Author Response · Authors · 2023-11-13
> **Rebuttal**
>
> - *On missing references*
>
> We are truly grateful to the reviewer for pointing out these very interesting references. We had indeed missed to what extent this part of the literature was relevant to our work. When we wrote the Related Work section, we wanted to focus on the specific context of (pre-)training models in a self-supervised fashion. Hence, we specifically cited concurrent works that used contrastive learning in that specific context.
>
> We still advocate that our work is novel *in that specific context*, but we will adjust some phrasings in our article to avoid conveying any impression of over-statement. In the next few days, we will upload a revised version of our paper with an updated Related Work section that engages better with the line of work you refer to.
>
> We would also like to underline **the differences between our work and the references the reviewer suggested**. A significant difference between our approach and importance sampling methods [1,2,4] is that our regularization is done on the hidden representation distribution and not on its support. As a matter of fact, we sample the negative samples uniformly from the batch, which means several instances of a given token may be found among them. The sorting objective of [3] is indeed an application of the NCE loss for the language modeling task, but they do not use the input embeddings as targets, resulting in a memory overhead. While [5] uses a tied input/output approach, they freeze the input embeddings and use trained static embeddings as targets. In [6], the authors also use various trained non-contextual embeddings to anchor their predictions. Although it could be interesting to see how the EBM approach from [7] works with our encoder, it is not clear how it could solve the problem of calibration in the decoder case, as it is used to sort sequences sampled from the pretrained LM, while our causal headless model needs to be adapted for token-level generation.
>
> **Questions**
>
> - *On better data-efficiency*
>
> We attempt to explain the better performance of our approach in **Appendix A**, as detailed in the General rebuttal. Our argument is also used by [4] (section 3.1.1), which we will mention accordingly.
>
> - *Supplementary compute*
>
> See *General rebuttal / On the additional cost of causal fine-tuning*. The additional fine-tuning costs 1,7% of the pre-training compute.

---

> ### Author Response · Authors · 2023-11-21
>
> Dear reviewer CHGU,
>
> We thank you again for your remarks and suggestions. We have integrated the missing literature in the updated version of the article, and we have rewritten sentences that you have rightfully pointed out as misleading.
> Considering that the discussion phase is coming to an end, we would be happy to receive your feedback and to answer any supplementary questions that you may have.
>
> Best regards,
> Submission 2843 Authors

---

> ### Comment · Reviewer_CHGU · 2023-11-22
> **Answer to Rebuttal**
>
> I appreciate the author's detailed response to my and other reviewer's comments. I better understand your goal and positioning with respect to the literature. However, I believe the current update made to the related works is lacking - as an enumeration of papers, with no real positioning - while your rebuttal here clearly position your work with respect to these papers. I also appreciate the added ablation study, and the hypothesis and visualization proposed in Appendix A.
> Overall, following these developments, I am raising my score to 6 (and would have liked to go in between 6 and 8, if the rating system allowed) - assuming that the authors would continue improving their related work section to provide accurate positioning.

---

> > ### Author Response · Authors · 2023-11-22
> >
> > We are grateful for your appreciation of our rebuttal and revisions. Thank you very much for updating your overall score.
> >
> > We tried our best to present the literature you mentioned in a way that implicitly showed the differences with our work. Furthermore, we included citations in relevant parts of the rest of the article, to better include the new references. However, we understand that the paragraph was not clear enough, and we thank you for the feedback.
> >
> > We revised the related works section to make the differences with our approach clearer:
> >
> > *The idea of relieving language models of the need to predict probabilities over the whole token vocabulary has been explored in the importance sampling literature [1,2,3,4]. These methods approximate the denominator of the softmax by using only a subset of the possible tokens. Those approaches usually rely on variants of the Noise-Contrastive Estimation objective [NCE paper] that use unique negative samples, **contrary to our approach that samples representations uniformly from the batch**. [5] and [6] use contrastive objectives based on cosine-similarity to match pre-trained static embeddings for Machine Translation. **We instead use the model's input embeddings as trainable target representations.***
> >
> > Please let us know if you prefer this new version, so that we can update the submission accordingly. We also notice that your presentation and contribution scores have remained unchanged. Could you please specify any concerns that persist in these areas?

---

### Official Review · Reviewer_ccZo · 2023-11-01

**Soundness:** 3 good
**Presentation:** 3 good
**Contribution:** 3 good
**Rating:** 6
**Confidence:** 4

**Summary:**

This paper proposes replacing the standard cross-entropy-based language modeling objective (classification over token vocabulary) with a "contrastive weight tying" objective, which is more memory and compute efficient during training. Computing the full cross-entropy loss requires loading and multiplying the final model embeddings by a $d_{model} \times V$ matrix, where $V$ is the vocabulary size (often $\geq 30k$), which can be quite expensive (especially for small models, with few layers and $d_{model} \ll V$). Instead, this paper proposes using a contrastive objective, where the targets are the input token embeddings for the masked tokens ("weight tying"), and the negative samples are taken from the other "masked" tokens. It shows in experiments that the proposed method is generally more efficient, and attains better performance, than the standard cross-entropy-based training. Lastly, to allow for generating text (using next token prediction) with these "headless" models, the paper proposes performing fine-tuning with a LM head for a small number of tokens ($< 2$% of pre-training dataset).

**Strengths:**

- The proposed method can reduce the number of trainable parameters meaningfully for small models with large vocabulary sizes (e.g., by my calculations, reduce # parameters of BERT-base by ~18%).
- The proposed method generally yields better accuracy on downstream tasks than standard LM training (across mono-lingual encoder experiments with BERT-base, mono-lingual decoder experiments with Pythia-70m, and multi-lingual encoder experiments with distilled multi-lingual BERT).
- This method decouples the training speed from the number of tokens in the vocabulary, which allows for choosing the vocabulary size that attains best performance.

**Weaknesses:**

- The proposed method is not very novel --- straightforward application of contrastive loss function to LM training.
- The experiments are relatively small-scale, only considering models < 140m parameters. Unclear if any of the benefits of this method hold for larger models.
- The paper does not discuss that for larger models, the $d_{model} \times V$ classification matrix corresponds to a tiny percentage of the total number of model parameters (and thus, total memory and computation during training). For example, it corresponds to roughly 2.5% and 0.4% of the Llama2 (7b) and Llama2 (70b) model parameters, respectively.
- The proposed method adds a step to decoder-only LM training, as the model must first be pre-trained in a "headless" way, and then fine-tuned with a LM classification head. This adds complexity to the training process.
- The paper does not perform careful ablations to help understand why the proposed method attains better model performance than standard cross-entropy (CE) LM training. For example, it could have considered weight-tying with the CE loss, or a contrastive loss without weight tying. This latter option could still be implemented in a compute-efficient way during training (since only a subset of columns of the classification matrix would need to be used). It could have also tried using different numbers of negative samples for the contrastive loss function (assuming # negative samples was decoupled from # masked samples), such that on one extreme (# neg samples = vocab size) the loss would be equivalent to the cross-entropy loss function (modulo the weight tying). Overall, I was confused by why the proposed method would yield better-performing models than cross-entropy.

Note that I was torn on whether to assign this paper "marginal accept" or "marginal reject", but chose marginal accept given that I see the efficiency benefit of this approach for smaller models, and was intrigued that this approach could improve model quality. The reasons I was considering marginal reject were that I felt the method was not very novel, only leads to meaningful efficiency gains for small models (and no experiments with medium/large models were performed), and because I felt the paper didn't do a good job explaining why the proposed method performs better. Open to being swayed.

**Questions:**

- Am I understanding correctly that for the monolingual decoder experiments, the proposed method offers no speed advantages, because  "negative samples correspond to every input embedding at a different position in the batch"? Would it make sense to decouple the number of negative samples from the number of "masked" samples, to allow for controlling these two hyperparameters separately?

---

> ### Author Response · Authors · 2023-11-13
> **Rebuttal**
>
> We sincerely thank the reviewer for its thorough review, and for its careful technical considerations.
>
> **Weaknesses**
> - Points 1, 2, 4 and 5 are addressed in the *General Rebuttal*
> - *Scaling of the input embeddings proportion*
>
> We entirely agree that as monolingual models scale, the benefits of our approach in terms of memory and compute efficiency will progressively decrease. However, we argue that the benefits of our approach in terms of data-efficiency, i.e. the final performance of a model trained on the same data, can be expected to scale. We addressed this point in **Appendix B**.
>
> Moreover, we believe that the computational benefits theoretically hold for large language models with larger vocabularies. For mGPT (1.3B parameters, 100000 tokens vocabulary) and Bloom-7B1 (250880 tokens vocabulary), input embeddings represent ~15% of the model parameters. If one used the XLM-V vocabulary (1M tokens) to train the architecture of Llama-7B, the input embeddings would represent ~60% of the trainable parameters. We advocate that these use cases could greatly benefit from our approach.
>
> - *On the lack of ablations*
>
> We thank the reviewer for this very relevant question. We addressed most of its concerns in *General Rebuttal / On ablation study*. As a clarification, we would like to point out that increasing the number of negative samples would not make the CWT loss equivalent to the cross-entropy loss function. The negative samples would indeed follow the Zipfian distribution of the data instead of the uniform distribution implicitly used in the cross-entropy loss. We hypothesize that regularizing based on the unigram distribution instead of the uniform vocabulary distribution may also explain the performance improvement of CWT.
>
> We also invite the reviewer to read **Appendix A** that gives some insight on the reasons of the effectiveness of our method.
>
> **Questions**
> - *On choosing the number of negative samples*
>
> The speed benefit of our method is indeed reduced in this specific setting, as we used a micro-batch size large enough to make the number of negative samples comparable with the vocabulary size. We still observed a training speed-up, which may be explained by optimization considerations (with fewer parameters to update) and/or hardware considerations.
>
> We mention the idea of considering different amounts of negative samples in a footnote (page 4), but we left the exploration of this idea for future work. Preliminary results indicate that we can indeed lower the number of negative samples without loss of performance, thus further increasing the training throughput.

---

> ### Author Response · Authors · 2023-11-21
>
> Dear reviewer ccZo,
>
> We thank you again for your remarks and questions. We have addressed your concerns on novelty, and conducted an ablation study similar to the one you suggested that was added to the paper.
> Considering that the discussion phase is coming to an end, we would be happy to receive your feedback and to answer any supplementary questions that you may have.
>
> Best regards,
> Submission 2843 Authors

---

> > ### Comment · Reviewer_ccZo · 2023-12-04
> > **Thank you for your responses**
> >
> > Thank you to the authors for their thoughtful replies to my comments and questions.
> >
> > Overall, I still feel unsatisfied by the ablation studies and the attempts to explain why the proposed method performs better than standard LM training (a question that *all* other reviewers raised). As mentioned in my review, I think there are many potential ablation studies that could help get to the root of this question, and the current paper only skims the surface. I believe the primary purpose of academic ML papers should be to teach/understand when/why certain methods are better than others (and the various trade-offs between them) as opposed to simply arguing for the superiority of a single method --- and I don't feel this paper does a good job in this regard. As such, I have chosen to not raise my score above a marginal accept.

---

### Official Review · Reviewer_A6os · 2023-11-01

**Soundness:** 3 good
**Presentation:** 3 good
**Contribution:** 3 good
**Rating:** 6
**Confidence:** 4

**Summary:**

This study introduces an innovative approach to self-supervised pre-training of language models. Instead of predicting probability distributions over token vocabularies, the method focuses on reconstructing input embeddings contrastively using Contrastive Weight Tying (CWT). This approach is applied to pretrain Headless Language Models in both monolingual and multilingual contexts, offering practical advantages. It is significantly more compute-efficient, data-efficient, and performant than classical predictive methods. The paper's contributions include the introduction of a new pretraining objective, the pretraining of encoder and decoder models for English and a multilingual encoder model, demonstrating the benefits of headless training, and exploring the effects of pretraining hyperparameters on downstream performance.

**Strengths:**

1. The paper maintains a high standard of quality. The methodology is well-structured and explained clearly, making it accessible to readers. The empirical results demonstrate substantial reductions in training computational requirements, improved downstream performance, and increased data efficiency. The study includes a significant increase in GLUE score and LAMBADA accuracy, indicating a quality improvement over classical LMs.

2. The paper is well-written and clear in its explanations. It effectively communicates the core concepts and methodologies involved in the proposed approach. The objectives and benefits of headless training are well-articulated, and the exploration of pretraining hyperparameters adds to the clarity of the study.

3. The paper's significance lies in its departure from traditional language model pre-training by introducing a contrastive objective. Sufficient experiments validate the advantages of the proposed contrastive learning loss over traditional cross-entropy loss, and the experimental results are relatively convincing.

In summary, it is a valuable contribution to the field of self-supervised language model pre-training. The proposed approach has the potential to impact the efficiency and effectiveness of language models in various applications.

**Weaknesses:**

1. The proposed method is not very novel and original. Although it departs from traditional probability prediction, offering a unique perspective on language model pre-training, there is lots of similar work that leverages the similar contrastive learning method. Thus, the use of Contrastive Weight Tying (CWT) in this context almost combines the existing ideas into the pretraining model, and it is not a very novel and creative methodology;

2. For the currently most popular CLMs, the proposed method still requires fine-tuning on a small amount of training data; otherwise, the performance would be significantly worse than the vanilla approach, adding complexity to the training process;

3. With the current trend of scaling up language models, it would be even more valuable to validate the proposed method on larger generative LLMs, such as ChatGLM or LLaMa.

**Questions:**

1. I'm curious about the experimental results in Table 2. I wonder why there is a significant improvement on all test sets except the CB test set. Why does the Headless model perform so much worse than the vanilla model on the CB test set? Could the authors provide a detailed analysis and explanation for this? Is there anything special about the CB test set?

2. In Table 3, why is there no output for the Headless model's perplexity (ppl) on the validation set?

3. I believe it's reasonable that the Headless model is more memory-efficient and exhibits higher training and inference efficiency on the XNLI benchmark compared to the Vanilla model. However, why is there such a significant improvement in quality as well? Where do these quality improvements primarily come from? Could the authors provide a more specific explanation?

---

> ### Author Response · Authors · 2023-11-13
> **Rebuttal**
>
> We are grateful to the reviewer for its clear and relevant review.
>
> **Weaknesses**
>
> 1, 2, 3 : see *General Rebuttal*
>
> **Questions**
>
> 1. *On the CB dataset*
>
> We analyzed the reason for this discrepancy more thoroughly. We found that the CB validation set we used for our analysis **only counts 56 examples**. While the standard deviation of the performance of each model is close to 1, making the score gap meaningful across seeds, it only requires a correct classification on two more samples to explain such a gap with this sample size. We believe that this explains the observed discrepancy.
>
> 2. *No validation perplexity in Table 3*
>
> We did not report perplexity for headless models during our initial training, as it would have required us adapting the forward procedure to use the transposed input embeddings matrix as a projection head during validation steps only. Moreover, we did not realize *before training* that we would need to put the model through a fine-tuning phase after pre-training. Finally, as shown by the evaluation on the Lambada dataset, the perplexity level of the headless model before fine-tuning is not really relevant for comparison with the other models.
>
> We would very much like to compute a perplexity score now to answer your question, but the OpenWebText2 dataset is not hosted anymore (https://huggingface.co/datasets/the_pile_openwebtext2/discussions/5), which does not let us do so.
>
> 3. see *General Rebuttal / On understanding the performance gap*

---

> ### Author Response · Authors · 2023-11-21
>
> Dear reviewer A6os,
>
> We thank you again for your remarks and questions. We hope to have addressed them with our rebuttal and the updated version of our article.
> Considering that the discussion phase is coming to an end, we would be happy to receive your feedback and to answer any supplementary questions that you may have.
>
> Best regards,
> Submission 2843 Authors

---

### Author Response · Authors · 2023-11-13
**General Rebuttal**

We thank all the reviewers for their very valuable feedback. We first address concerns that were shared by several reviewers:

## On the lack of novelty
We are well aware that our CWT objective function is not novel nor creative as a learning objective with respect to general ML. What we wish to stress out in this article is that our work distinguishes itself from closely related literature by exploring a contrastive objective as a replacement for cross-entropy **in the context of pre-training token-level language models** in a self-supervised fashion. Recent concurrent work in this specific domain usually apply contrastive objectives as auxiliary losses that are added to cross-entropy (e.g. https://arxiv.org/pdf/2111.04198.pdf or https://arxiv.org/pdf/2202.06417.pdf). We believe our work is novel *in that specific line of work*. We will rephrase some statements made in the paper to clarify this point in an updated version.

## On the additional cost of causal fine-tuning
We agree that the necessary fine-tuning of the LM classification head slightly complexifies the pre-training process from a technical/coding perspective. However, we only needed to use a **small fraction of the training compute (1.7%)** to obtain a performant language model. In Figure 5a and 5b, we include this overhead on the x-axis for the fine-tuned model, to demonstrate that the added computation complexity is actually negligible.

## On understanding the performance gap
We attempt to answer this question in **Appendix A**. To summarize, we argue that cross-entropy pushes semantically close words apart at each training step, by incidentally using them as negative samples. Our CWT objective instead pushes close words apart only if they appear in the same batch, which is less likely. It results in a slightly higher average cosine-similarity for synonyms for headless models, which indicates more meaningful representations.

Another argument that is not directly mentioned in our paper is that our objective addresses the problem of updating rare and unused token embeddings at each step, which is a flaw of the cross-entropy loss (https://arxiv.org/abs/1907.12009).

## On scale
We mention the limitations related with model scale in **Appendix B**. This paper was written in a context of limited computation capabilities, and extending a new pre-training approach to larger scales is extremely costly in terms of compute. Validating our method on Llama or ChatGLM, as suggested by Reviewer A6os, would have implied pre-training such models from scratch, which is inaccessible for our institution. For instance, the smallest Llama-2 model (7B parameters) took 184k A100 GPU hours to pretrain, which can be roughly estimated to cost ~200k$.

## On ablation study
We want to clarify that we used classical weight-tying in all our “vanilla” models trained with cross-entropy, as we believe it is now standard in the literature. With regard to the other ablation Reviewer ccZo suggests, i.e. using an additional static embedding matrix for negative samples apart from the one used for input, we did not run this experiment as it results in adding a substantial count of parameters to the model, which could have biased the downstream performance comparison. Nevertheless, we are currently running the corresponding experiment at small scale, and we will communicate the results as soon as possible.

About trying different numbers of negative samples (as suggested by Reviewer ccZo), we think this idea is worth exploring, as we mentioned in the footnote of page 4. At submission time, we decided to leave it for future work due to space limitations, as it would have required a hyperparameter search. We now have preliminary results that indicate that it is possible to select fewer negative examples without loss of performance, up to a certain point.

---

### Author Response · Authors · 2023-11-17
**Revision**

Based upon the numerous comments and suggestions made by the reviewers, we updated our paper in several aspects:
- **Ablation study** : We added an ablation study for small-scale models, where we compare our approach with two variants (inspired from reviewer ccZo) : one where the loss is the addition of cross-entropy and CWT, and one contrastive-only approach that does not use weight-tying, i.e. that uses external static embeddings as targets (NCE wo/ WT). Here are the results, that can also be found on page 9 :

|       Objective      | Parameters | Throughput | GLUE avg. |
|:-------------------:|:----------:|:----------:|:---------:|
|    Cross-Entropy    |     x1     |     x1     |   82.45   |
| CWT + Cross-Entropy |     x1     |    x0.87   |   82.93   |
|     NCE (wo/ WT)    |    x1.57   |    **x2.47**   |   82.91   |
|         CWT         |     x1     |    **x2.13**   |   **83.37**   |


- **New references** : We updated the related work section (pages 2 and 3) by including most of the references suggested by revierwer CHGU. We contextualized our claims of novelty, and we also cited the appropriate papers along our article.

- **Extended Appendix A (explanation for performance gain)** : We tried to provide a deeper intuition about the data-efficiency gain of our method (see page 14).

We thank again all the reviewers for their suggestions that add great value and completeness to our article. We are open to additional suggestions and comments before the end of the discussion period.

---

### Author Response · Authors · 2023-11-23
**Reminder: 4 hours left in the discussion period**

Dear reviewers,

The reviewer-author discussion period will come to a conclusion **in under 4 hours**. If you have any further thoughts or comments, don't hesitate to share them with us. We remind you that we conducted additional work following your reviews, and we uploaded a revised version of the article. We sincerely thank you for your dedication and time in reviewing our submission.

Warmest regards,
Author(s)

---

### Meta-Review · Area_Chair_QQoD · 2023-12-04

**Metareview:**

This paper proposes replacing the standard cross-entropy-based (large) language modeling objective (classification over token vocabulary) with a "contrastive weight tying" objective, which is more memory and compute efficient during training. Computing the full cross-entropy loss requires loading and multiplying the final model embeddings by a matrix, which can be quite expensive (especially for small models, with few layers, a small dimensionality and a large vocabulary). Instead, this paper proposes using a contrastive objective, where the targets are the input token embeddings for the masked tokens ("weight tying"), and the negative samples are taken from the other "masked" tokens. It shows in experiments that the proposed method is generally more efficient, and attains better performance, than the standard cross-entropy-based training. Lastly, to allow for generating text (using next token prediction) with these "headless" models, the paper proposes performing fine-tuning with a LM head for a small number of tokens (<2% of pre-training dataset).

Strengths
- The proposed method can reduce the number of trainable parameters meaningfully for small models with large vocabulary sizes (e.g., by my calculations, reduce # parameters of BERT-base by ~18%).
- The proposed method generally yields better accuracy on downstream tasks than standard LM training (across mono-lingual encoder experiments with BERT-base, mono-lingual decoder experiments with Pythia-70m, and multi-lingual encoder experiments with distilled multi-lingual BERT).
- This method decouples the training speed from the number of tokens in the vocabulary, which allows for choosing the vocabulary size that attains best performance.

Weaknesses
- The proposed method is not very novel --- straightforward application of contrastive loss function to LM training.
- The experiments are relatively small-scale, only considering models < 140m parameters. Unclear if any of the benefits of this method hold for larger models. The paper does not discuss that for larger models, the  classification matrix corresponds to a tiny percentage of the total number of model parameters (and thus, total memory and computation during training). For example, it corresponds to roughly 2.5% and 0.4% of the Llama2 (7b) and Llama2 (70b) model parameters, respectively.
- The paper does not perform careful ablations to help understand why the proposed method attains better model performance than standard cross-entropy (CE) LM training (although some were presented during the discussion)

**Justification For Why Not Higher Score:**

The paper leaves open some questions around applicability to larger data/ models, which would be required for acceptance at a higher level of certainty.

**Justification For Why Not Lower Score:**

The paper presents meaningful results and after discussion ,all reviewers recommended acceptance. The paper does not have any obvious flaws that authors have not been able to address in discussion.

---

### Decision · Program_Chairs · 2024-01-16

Accept (poster)